# Clinical outcomes of corneal neurotization using sural nerve graft in neurotrophic keratopathy

Manu Saini[1☯], Aman Kalia[1☯], Arun K. Jain[1]*, Sunil Gaba[2☯], Chintan Malhotra[1‡], Amit Gupta[1‡], Tanvi Soni[1‡], Kulbhushan Saini[3‡], Parul Chawla Gupta[1‡], Manpreet Singh[1‡]

**1** Advanced Eye Centre, Post Graduate Institute of Medical Education and Research, Chandigarh, India, **2** Department of Plastic Surgery, Post Graduate Institute of Medical Education and Research, Chandigarh, India, **3** Department of Anaesthesia, Post Graduate Institute of Medical Education and Research, Chandigarh, India

☯ These authors contributed equally to this work.
‡ CM, AG, TS, KS, PCG and MS also contributed equally to this work.
* aronkjain@yahoo.com

**Data Availability Statement:** All relevant data is available in the Supporting Information file named "S1 Table".

## Abstract

### Objective

To evaluate the efficacy of corneal neurotisation using sural nerve graft coaptation of the contralateral supratrochlear nerve in unilateral neurotrophic keratopathy and corneal anesthesia. Corneal neuralization has emerged as a potential option in the treatment of neurotropic keratopathy, however not free from the predicament. We evaluated the long-term outcome of corneal neurotisation in the treatment of unresponsive unilateral neurotropic keratopathy using surgical variations to mimic and expedient the surgical procedure.

### Methods

A Prospective interventional study involving patients with unilateral neurotrophic keratopathy (NK) who did not respond to medical measures was conducted. The study parameters evaluated were best-corrected visual acuity improvement, ocular surface evaluation parameters [tear break-up time (TBUT), Schirmer's 1, and ocular surface staining scores (corneal and conjunctival staining)], central corneal sensation (Cochet Bonnet esthesiometer), subbasal nerve fiber length (SBNFL), and sub-basal nerve fiber density (SBNFD) determined by central confocal microscopy at recruitment and during follow-up at 1-month, 3-month, 6-month, 9-month and 12-month respectively, following corneal neurotization.

### Results

Eleven eyes of 11 patients with unilateral neurotrophic keratopathy (NK) who underwent corneal neurotisation were studied. The mean follow-up was 10.09±2.31months (range, 6–12). Mean best corrected visual acuity in log MAR at baseline, 1.35±0.52 improved significantly to 1.06±0.76 (P = 0.012) at 3 months and continued to 0.55±0.60 (P = 0.027) at 12 months. There was a significant reduction in NK grade severity and improvement in the ocular surface as early as 1 month, and central corneal sensations (P = 0.024) as soon as 3

**Funding:** The author(s) received no specific funding for this work.

**Competing interests:** The authors have declared that no competing interests exist.

months. Mean corneal SBNF improved from 3.12±1.84 mm/mm2 to 4.49±1.88 at 1 month (P = 0.008), 13.31±3.61 mm/mm2 (P = 0.028) at 12 months. Mean central corneal SBNFD evident at 6 months was 1.83±2.54no/mm2 (P = 0.018) and 4.90±3.12no/mm2 (P = 0.028) at 12 months.

## Conclusion

This study substantiates the routine practice of corneal neurotisation by simplifying the intricacies observed during the procedure.

## Introduction

Neurotrophic keratopathy (NK) is a degenerative corneal condition that originates from damage to the trigeminal innervation and serves as an integral component of ocular surface homeostasis, blink reflex, corneal wound healing, tear production, and normal limbal stem cell function [1, 2]. Sensory nerves secrete many neuropeptides, such as substance P, calcitonin gene-related peptide, and cholecystokinin, which are involved in trophic effects and their depletion induced by capsaicin, on culture in new-born mice evidenced neuroparalytic-like corneal alterations [3]. Human corneal epithelial cells and keratocytes influence the survival, differentiation, and maturation of nerve fibers by releasing neuropeptides, neurotrophins, and growth factors such as nerve growth factors. This symbiotic relationship is essential to maintain the integrity and function of the ocular surface, therefore, it is imperative to restore the corneal sensation [4].

Neurotrophic keratopathy is the most intricate corneal condition that involves a stepwise therapeutic approach guided by the severity of the disease. All conventional measures are aimed at protecting the corneal surface and averting its epithelial breakdown [5]. Recombinant nerve growth factor (Cenegermin, Dompè Farmaceutici, Milan, Italy) emerges as much greeted advent in the management of NK and bestows a paradigm shift in medical management by addressing the root pathology [6]. However, its efficacy in clinical trials [7, 8], high costs, inability to address the underlying absence of corneal innervation and subsequent perpetuation of progressive vision loss confine its widespread use in routine clinical practice. Therefore, Corneal neurotization (CN) has evolved as a potential therapeutic surgical technique, being able to re-establish the corneal sensation and reverse corneal alteration by providing a functional source of innervation [9]. This can be done directly by transposing the adjacent healthy nerve or indirectly by using nerve grafts coapted to the healthy donor nerve [10, 11]. Both surgical techniques were attributed to the improvement in corneal sensations, notably 80% for direct and 83.3% success rate for indirect CN at one year [12].

However, there are clinical questions to ponder, especially in the context of longer surgical duration, intricacies, multidisciplinary team involvement, and infrastructure facilities that preclude its frequent acceptance in routine practice. Hence, it provides a great impetus to appraise the outcomes of corneal neurotisation in therapeutically non-responding unilateral NK. Because of the heterogeneous etiology of NK in the recruited eyes, we preferred sural nerve graft coaptation to the contralateral supratrochlear nerve acting as a donor nerve to avoid the use of a possibly involved ipsilateral donor nerve.

## Methods

### Study design and participants

A prospective interventional Corneal neurotization study using sural nerve graft coaptation to the contralateral supratrochlear nerve in 11 eyes of 11 patients with unilateral neurotrophic

keratopathy, non-responding to the medical measures was performed. The patients were recruited from the period February 2021 to December 2021 from the cornea clinic and followed from February 2021 to June 2022. The study conforms to the principles of the Helsinki Declaration and has received approval from the Institute Ethics Committee at the Postgraduate Institute of Medical Education and Research, Chandigarh, dated January 25,2021, with approval number INT/IEC/2021/SPL-161. The authors also confirm that the study has been registered under Clinical Trial Registry, India with Reference no- CTRI/2021/10/037280. Neurotrophic keratopathy that was caused by viral keratitis, impairment of the trigeminal nerve following intracranial space-occupying lesions such as acoustic neuroma, the damaged ophthalmic branch of the trigeminal nerve following neurosurgical events in patients with age more than 18 years and willingness to follow up were enrolled in the study. All the recruited eyes were on preservative-free artificial tears in the form of drops and ointment at all stages of disease severity for at least 6 months before enrolment. In addition, systemic oral tetracycline in a dose of 100mg BD for two weeks was considered initially at diagnosis for stage 2 and 3 NK grades before proclaiming them medically non-respondent eyes. The use of topical antibiotic eye drops to prevent infection at NK grades 2 and 3 was considered intermittently. In the case of herpes etiology NK, patients did not have a recurrence in the past six months, before enrolment.

Eyes with associated lid malposition, prior corneal surgery, history of diabetes, leprosy or peripheral neuropathy and incomplete follow-up were excluded.

Demographic data comprises age, gender, the affected eye, diagnostic aetiology and duration of the disease recorded. Comprehensive ocular examination was performed in all the recruited eyes including best-corrected Snellen's visual acuity, slit lamp biomicroscopic examination of the ocular adnexa, anterior segment evaluation and posterior segment examination using +90Dioptre lens or B scan ultrasound in the presence of media haze obscuring posterior segment visualization. Schirmer's 1 test [13] for tear production, fluorescein tear break-up time {TBUT} [13] for tear film stability, National Eye Institute ocular surface staining scores [13] of cornea and conjunctiva describing ocular surface impairment and neurotrophic keratopathy grade determined by Mickie's classification [14] were recorded. The flowchart of the study was shown in Fig 1.

**Corneal sensation measurement.** The central and peripheral (superior, inferior, nasal, and temporal) corneal sensation thresholds were evaluated using a Cochet Bonnet Aesthesiometer (CBA; Luneau, Paris, France) by direct contact, which stimulates the corneal nerves. The longest filament (6 cm; 0.12 mm diameter) corresponding to the lowest threshold was applied gently against the anterior corneal surface. Continual stimulation was provided by reducing the filament length in 0.5 mm steps until the stimulus was felt. The corneal sensation threshold recorded was the measured filament length (cm), which provided a 50% positive response from four stimuli presentations [15].

**Central corneal in vivo scanning slit confocal microscopy analysis.** Confocal microscopy was performed in all eyes using the Heidelberg Retinal Tomograph with a Rostock corneal module (HRT3-RCM, Dossenhein, Germany). Central corneal images were obtained in manual gain mode using the standard setting of a 63X objective lens, utilizing a 670 nm red wavelength helium-neon diode laser as an illumination source. The three best images for each eye were selected for the analysis. Among the selected images, the best one, comprising the maximum number of nerves at the subbasal plexus level, was designated for analysis.

The subbasal nerve fiber layer was characterized by unmyelinated nerve fiber bundles consisting of straight and beaded fibers that course in the basal aspect of the basal epithelial cell layer on confocal microscopy. Subbasal nerve fiber length (SBNFL) image analysis was performed using freely downloadable ACC Metrics software(https://weillcornell.az1.qualtrics.

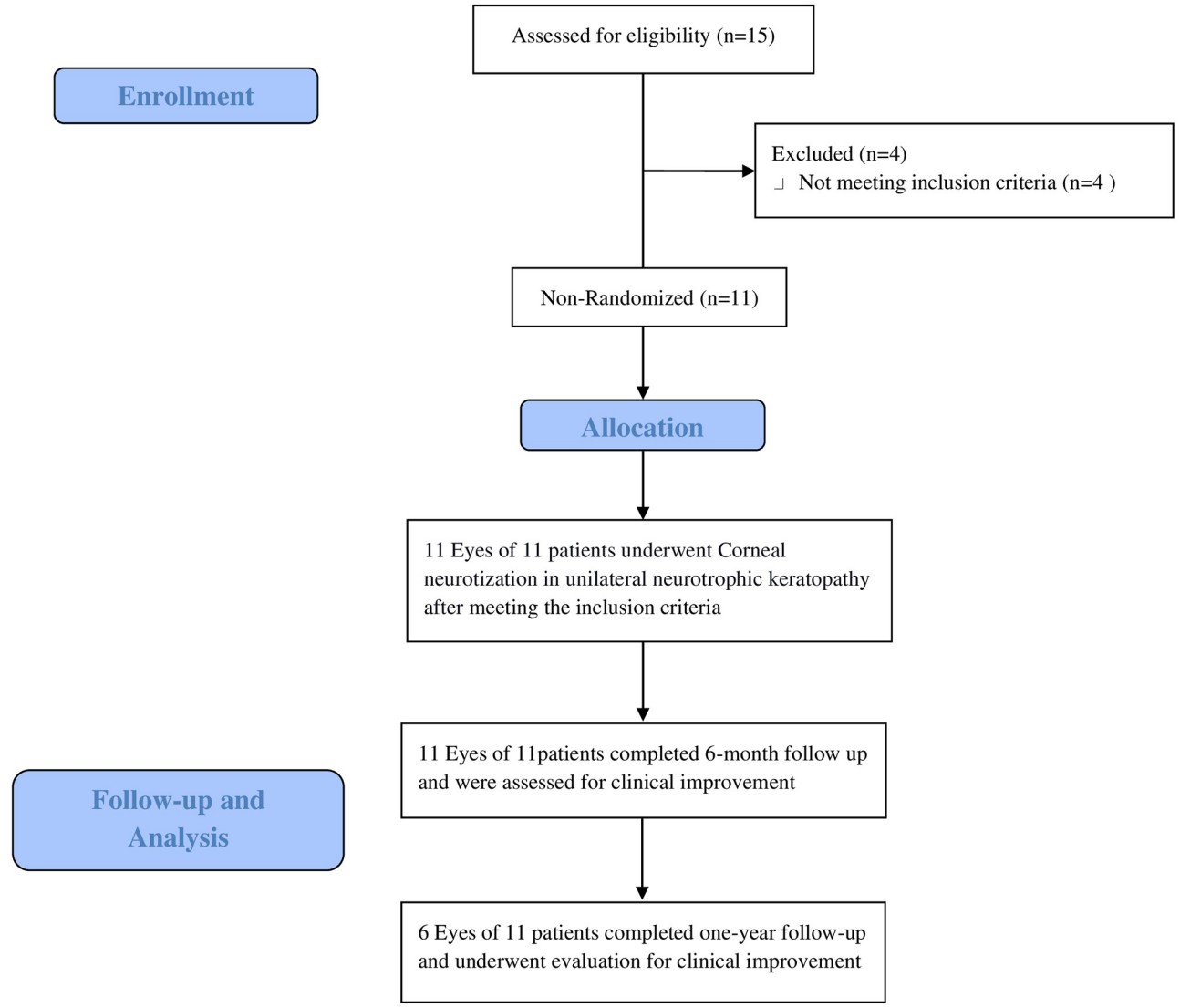

**Fig 1. Flowchart of enrolled eyes.** Flowchart of patients recruitment, follow-up and analysis who underwent corneal neurotization using sural nerve graft coaptation to the contralateral supratrochlear nerve in unilateral neurotrophic keratopathy, non-responding to the medical measures.

com/jfe/form/SV_6o2ji0suM4jQinb). ACC Metrics V.2 automatically analyzed the central confocal microscopy images with a field of view of 400×400 mm$^2$ obtained using the Heidelberg HRT III corneal confocal microscope and quantified the nerve fiber measurements, namely subbasal nerve fiber density (the total number of major nerves per millimeter squared of corneal tissue) and subbasal nerve fiber length (the total length of all nerve fibers and branches within the area of corneal tissue) from single or multiple corneal confocal microscopy images.

**Surgical procedure.** The sural nerve, acting as an interpositional graft, was used for corneal neurotization, as described by Elbaz et al [10]. in all recruited eyes with unilateral NK. A Multidisciplinary team of a plastic surgeon (SG) and ophthalmologist (MS) performed the procedure in all recruited eyes (Fig 2). A longitudinal incision was made approximately 2 cm posterior and 2–3 cm proximal to the lateral malleolus to identify the sural nerve where

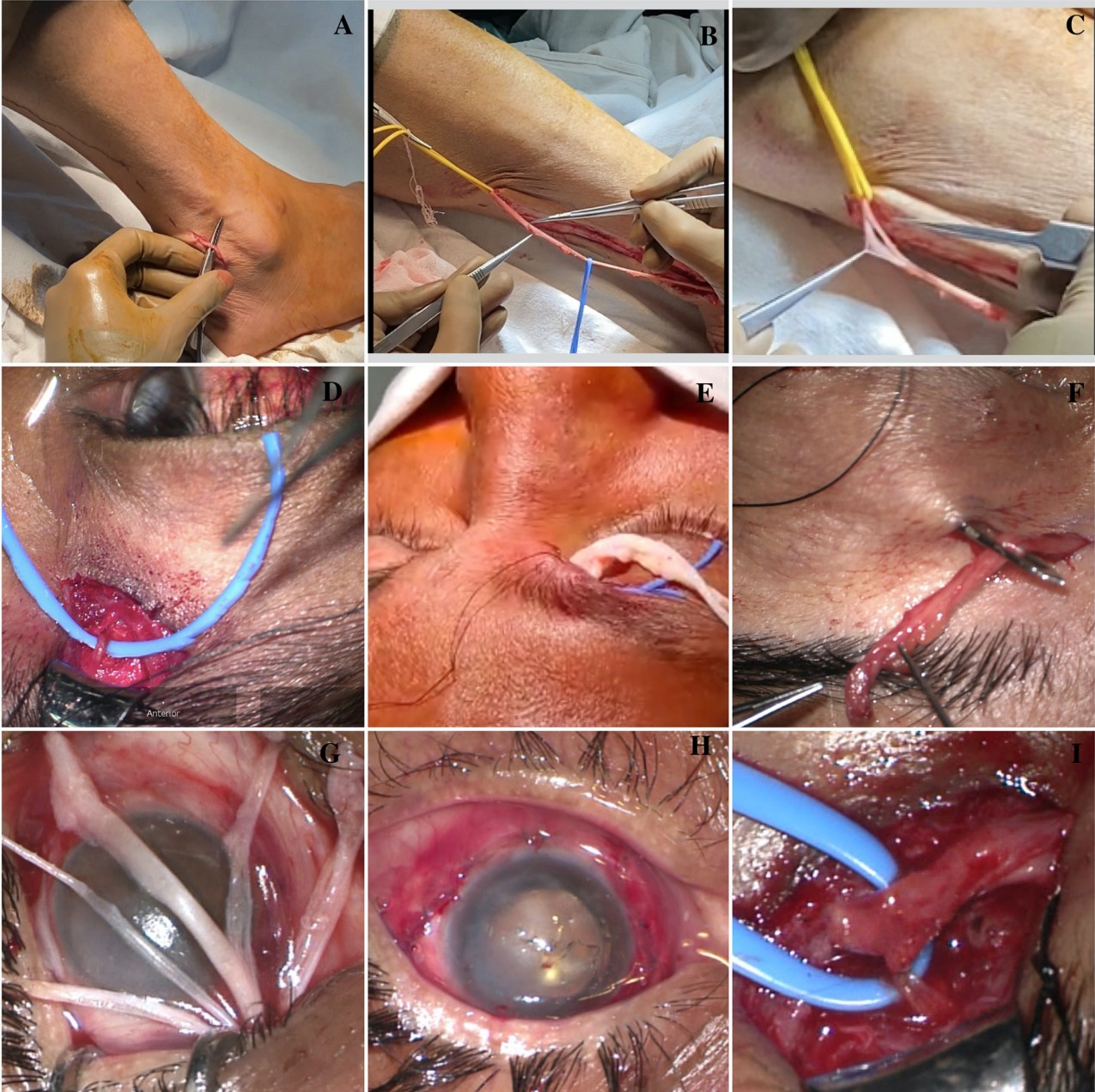

**Fig 2. Surgical steps of corneal neurotization.** (A) Identification and exposure of sural nerve, (B) Desired length of 10-12cm of the sural nerve was traced distally. (C) Careful fascicle separation through the epineurium window at the distal end. (D) Donor supra-trochlear nerve identification, (E) Reverse sural nerve tunnelling across the nasal bridge and (F) courses through sub-brow blepharotomy incision. (G) Fine segregation and recognition of separated fascicles under a microscope. (H) Secure individual fascicle sub-conjunctively at the perilimbal region, (I) End-to-side Coaptation of donor supra-trochlear nerve to inter-positional sural nerve graft.

significant branching of the nerve was not anticipated. The nerve was traced upward to obtain the desired length of 10–12 cm. Fascicles were separated by blunt dissection for a short distance at the distal end by creating a window in the epineurium, keeping the nerve under gentle traction and in the attached position itself. Fascicle separation before severing the sural nerve is infrequently performed. This facilitates smooth separation, mitigates possible collateral

damage, and saves surgical time. The desired length of the sural nerve was cut at the proximal and distal ends and placed on a moist gauze piece.

At the same time, the supratrochlear nerve acting as the donor's nerve was accessed on the unaffected side using a transverse sub-brow incision extending at the level of the medial canthus to the medial limbus. The supratrochlear nerve passes beneath, and medial to, the supraorbital notch. It was accessed by tracing its branches to the confluence point, and the dissection was directed downwards at the periosteum and towards the medial canthus to expedite its consolidated identification and confirmed by neurovascular bundle visualization. Fascicle separation of the graft and consolidated identification of the donor supratrochlear nerve was the most time-consuming steps of the corneal neurotization procedure and were hastened by the above-mentioned practice.

The reversed sural nerve was then tunnelled over the nasal bridge to the affected side and coursed through a sub-brow incision into the superior conjunctival fornix onto the bulbar conjunctiva. The epineurium at the distal end of the sural nerve was separated under a microscope with fine dissection, and the separated fascicles were identified. Five segregated fascicles were secured using 10–0 nylon and fibrin glue at the limbus by passing through the subconjunctival space. End-to-side coaptation of the sural nerve graft with the donor supratrochlear nerve was achieved by creating an epineural window on the supratrochlear nerve and secured with fibrin glue and 10–0 nylon sutures. The skin was closed using interrupted 6–0 silk sutures. Postoperative systemic antibiotics for seven days, topical steroid drops in the tapering dose, topical antibiotics, lubricating eye drops, and ointment were prescribed for 6 weeks in all recruited eyes with an antiviral drug in therapeutic dose till the patient was on the topical steroids (6weeks), followed by prophylactic antiviral dose for a period of total six months was prescribed to prevent reactivation of viral keratitis in viral etiology NK.

**Follow-up visits.** All data were recorded by a blinded evaluator using a predesigned proforma, who did not have access to identifying the recruited participants and patients were called for their follow-up visits at 1 week, 1 month, 3-month, 6-month, 9-month and 12-month. At each follow-up, best-corrected visual acuity, ocular surface evaluation [Tear film break up time (TBUT), Schirmer I test, and ocular surface staining score], corneal clarity as per Mackie's classification of neurotrophic keratopathy, central and four peripheral corneal sensations, and in vivo scanning slit confocal microscopy of the central cornea were measured as primary outcomes. The post-operative results will be compared with the non-affected (normal) eye to determine how much normalization has been achieved by performing the procedure. Any complications related to the intervention or surgical procedure in the recruited eyes were noted as the secondary outcomes.

## Statistical analysis

**Sample size.** Eleven patients of unilateral neurotropic keratopathy were determined for the study based on 1.6/10,000 prevalence of NK from the epidemiological data on conditions associated with NK- herpetic keratitis and post-surgical procedures [14]. 10% margin of error was preferred for sample calculation.

The following formula was used for the sample size calculation **n = N\*X / (X + N– 1),** where, $X = Z_{\alpha/2}{}^2 *p*(1-p) / MOE^2$, and $Z_{\alpha/2}$ is the critical value of the normal distribution at α/2 (e.g. for a confidence level of 99%, and α is 0.0), MOE is the margin of error, p is the sample proportion, and N is the population size

Quantitative variables are expressed as mean ± SD. Our data did not exhibit a normal distribution as assessed by the Shapiro–Wilk test, therefore non-parametric tests were performed for analysis. Ocular surface evaluation parameters comprising BCVA, TBUT, Schirmer's 1,

corneal, conjunctival staining scores and NK grade were analysed and compared at defined follow-up using the Wilcoxon signed ranks test. Similarly, corneal sensation and subbasal nerve plexus improvement after surgical intervention were compared with preoperative data at the destined follow-up, using Wilcoxon signed ranks test. Spearman's rho correlation coefficient measure was used to assess the correlation between age/duration and corneal sensation/SBNF plexus. Central corneal sensation and SBNFD appraisal at 6 months based on etiology and NK severity grade were determined using the Mann-Whitney U test. The SPSS version 20 program (SPSS Inc., Chicago, IL, US) was used for statistical analysis, with significance set at $p < 0.05$.

## Results

The demographic and clinical data of the 11 patients included in this study are shown in Table 1. The average duration of the underlying disease was 2.64±0.67 years (range, 2–4 years). There were six patients (54.54%) with NK stage II and five (45.5%) with NK stage III. All patients (100%) had completed a 6-month follow-up, 9/11(81.81%) had completed 9 months, and 6/11 (54.54%) had completed a 12-month follow-up visit following the procedure.

Ocular surface parameters comprising TBUT, Schirmer's 1 test, and corneal and conjunctival staining scores showed statistically significant improvement after the surgical intervention, notably as early as 1 month, and over each follow-up visit (Table 2). The recuperation was subsequently associated with a significant reduction in NK grade severity. Healing of the ocular surface emulated a significant improvement in best-corrected visual acuity from 1.35±0.52 (baseline) to 1.06±0.76 at 3 months (P = 0.012). Significant improvement was observed at each sequential follow-up until the last visit (Table 2).

Upon further analysis of the enrolled patients who completed 12 months of follow-up, two patients (33.33%) attained 20/20 Snellen's visual acuity, as shown in Figs 3 and 4.

Improvement in corneal sensation was statistically significant in all quadrants compared to preoperative value at follow-up visits that continued 1 year (P<0.05) (Table 3). It was

**Table 1. Demographic characteristics of the eyes with neurotrophic keratopathy, underwent corneal neurotization using sural nerve graft in the study.**

| Demographic data | Neurotrophic keratopathy eyes |
|---|---|
| No of eyes | 11 |
| Gender (M/F) | 7/4 (63.6%/36.4%) |
| Age mean±SD, (range) | 44.55±21.38, (25-74years) |
| Affected eye | |
| Right eye | 8 (72.7%) |
| Left eye | 3 (27.3%) |
| Diagnosis/NK grade [mean±SD] | |
| HSV | 6 (54.5%) [2.5±0.5] |
| HZO | 1 (9.1%) [1±0] |
| Facial nerve palsy | 4 (36.4%) [2.75±0.433] |
| NK grade | |
| Grade 1 | 0 |
| Grade 2 | 6 (54.54%) |
| Grade 3 | 5 (45.5%) |
| Duration of aetiology mean±SD, (range) | 2.64±0.67, (2–4 years) |

Data are presented as no. (%) or mean ± standard deviation unless otherwise indicated

**Table 2. Result of ocular surface evaluation tests (mean±SD) following corneal neurotization using sural nerve graft in eyes with neurotrophic keratopathy.**

| Outcomes | No of eyes | BCVA | TBUT (s) | Schirmer's 1 test (mm) | Cornea staining scores | Conjunctival staining scores | NK grade |
|---|---|---|---|---|---|---|---|
| Pre-operative | 11 | 1.35±0.52 | 3.00±1.94 | 5.45±3.35 | 8.18±2.44 | 6.64±2.54 | 2.45±0.522 |
| Post-operative | | | | | | | |
| 1-month | 11 | 1.22±0.68 | 4.55±2.42 | 7.36±3.72 | 7.18±2.99 | 5.09±2.21 | 2.36±0.50 |
| 3-month | 11 | 1.06±0.76 | 5.36±2.42 | 8.27±3.69 | 3.73±2.00 | 2.18±1.16 | 1.64±0.80 |
| 6-month | 11 | 1.03±0.76 | 6.18±2.71 | 9.00±3.46 | 1.91±1.37 | 0.91±0.70 | 1.64±0.92 |
| 9-month | 9 | 0.94±0.76 | 7.44±2.65 | 10.11±3.72 | 1.33±1.32 | 0.67±0.50 | 1.44±0.52 |
| 1year | 6 | 0.55±0.60 | 9.50±2.81 | 12.83±3.92 | 0.50±0.54 | 0.33±0.51 | 1.00±0.63 |
| P-value | | 0.030 | 0.001 | 0.001 | 0.001 | 0.001 | 0.001 |
| Pre vs 1-month | | 0.172 | 0.007 | 0.017 | 0.016 | 0.011 | 0.317 |
| Pre vs 3-month | | 0.012 | 0.012 | 0.016 | 0.003 | 0.003 | 0.007 |
| Pre vs 6-month | | 0.012 | 0.005 | 0.005 | 0.003 | 0.003 | 0.007 |
| Pre vs 9-month | | 0.017 | 0.007 | 0.012 | 0.008 | 0.007 | 0.003 |
| Pre vs 1 year | | 0.027 | 0.027 | 0.027 | 0.026 | 0.027 | 0.020 |

demonstrated earliest in 8/11 (72.72%) patients at the 3-month follow-up and in 9/11 (81.81%) patients at the 6-month follow-up. However, two patients who had completed a 6-month follow-up failed to express improvement in corneal sensation.

Preoperatively, in all the recruited eyes subbasal nerve fiber density was not appreciated and therefore unable to calculate, however, the baseline SBNFL 3.12±1.84 was calculated. The values of SBNFL≤ 14.4mm/mm2 and SBNFD ≤ 14.7 no/mm2, calculated using ACC Metrics V.2 software, were considered abnormal [16]. Patients who had completed the 1-year follow-up showed a significant increase in SBNFD (P = 0.028) and SBNFL (P = 0.028). Moreover, the earliest significant improvement in SBNFD (P = 0.018) was detected at the 6-month follow-up in the 7/11 patients (63.63%). Nevertheless, the most significant increase in SBNFL (P = 0.00) was observed at the 1-month follow-up in 9/11 patients (81.81%) (Table 3). The 95%

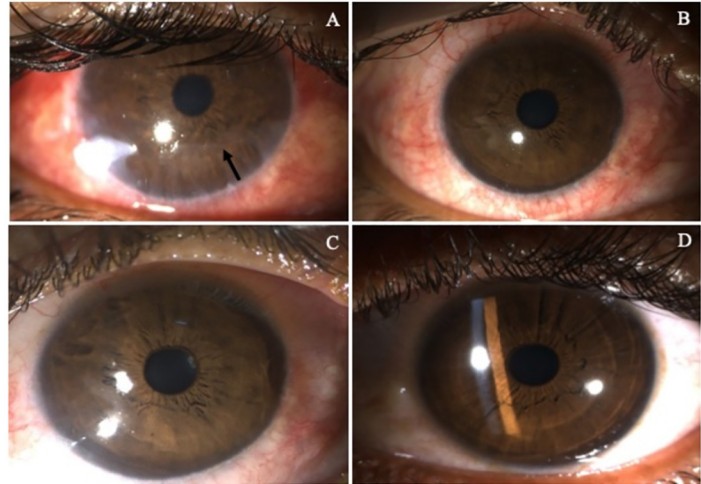

**Fig 3. Slit lamp photograph of the cornea, before and after corneal neurotization at sequential follow-up.** (A) Pre-operative clinical photograph showing NK grade 2 with inferior epithelial defect (arrow) and stromal oedema because of trigeminal and facial nerve involvement following brain tumour surgery. (B) One month postoperative photograph showed a marked reduction in epithelial defect and stromal oedema. (C) The six-month postoperative photograph revealed an increase in optical clarity with (D) Snellen best corrected visual acuity 20/20 at the 12-month follow-up.

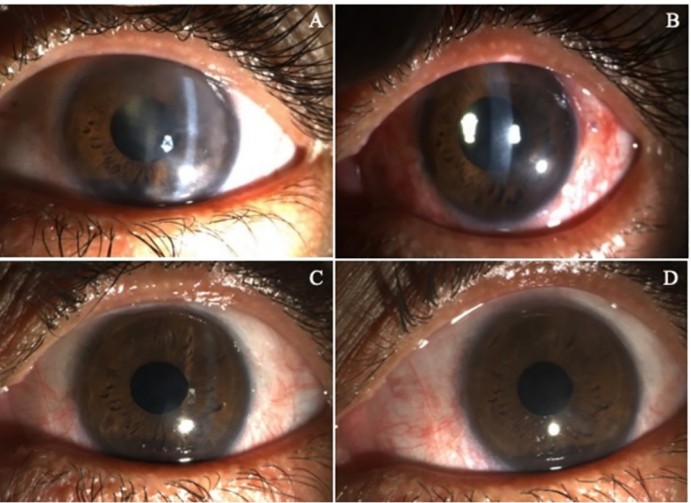

**Fig 4. Representative slit lamp photograph of NK grade 2 owing to herpes simplex infection.** (A) Pre-operative clinical photograph demonstrating stromal haze and oedema at baseline. (B) One month postoperative photograph showed a significant reduction in stromal oedema and haze. (C) Six-month clinical photograph represents an escalation in corneal transparency and (D) achievement of 20/20 Snellen's best-corrected vision at 12 months postoperative.

**Table 3. Clinical outcome of corneal neurotization [Mean±SD/Median (IQR)] on central and peripheral corneal sensation and central corneal sub-basal nerve fibre plexus in eyes with neurotrophic keratopathy.**

| Outcomes | No of eyes | CORNEAL SENSATIONS | | | | | SBNFD no/mm$^2$ | SBNFL mm/mm$^2$ |
|---|---|---|---|---|---|---|---|---|
| | | Central | Superior | Inferior | Nasal | Temporal | | |
| Pre-operative | 11 | 0.045±0.15 | 0.045±0.15 | 0.091±0.20 | 0.045±0.15 | 0.045±0.15 | 0.00±0.00 | 3.12±1.84 |
| Median (IQR) | | 0.00 (0.00–0.00) | 0.00 (0.00–0.00) | 0.00 (0.00–0.00) | 0.00 (0.00–0.00) | 0.00 (0.00–0.00) | 0.00 (0.00–0.00) | 2.79(2.00–4.53) |
| Post-operative | | | | | | | | |
| 1-month | 11 | 0.045±0.15 | 0.045±0.15 | 0.091±0.20 | 0.091±0.20 | 0.045±0.15 | 0.00±0.00 | 4.49±1.88 |
| Median (IQR) | | 0.00 (0.00–0.00) | 0.00 (0.00–0.00) | 0.00 (0.00–0.00) | 0.00 (0.00–0.00) | 0.00 (0.00–0.00) | 0.00 (0.00–0.00) | 4.48(2.81–6.24) |
| 3-month | 11 | 0.59±0.86 | 0.63±0.83 | 0.81±1.12 | 0.81±1.14 | 0.95±1.31 | 0.58±1.64 | 6.26±2.58 |
| Median (IQR) | | 0.50 (0.00–0.50) | 0.50 (0.00–0.50) | 0.50(0.00–1.00) | 0.50 (0.00–1.00) | 0.50 (0.00–1.00) | 0.00 (0.00–0.00) | 5.63 (4.00–8.23) |
| 6-month | 11 | 1.09±1.48 | 1.54±1.54 | 1.50±1.56 | 1.50±1.48 | 1.72±1.61 | 1.83±2.54 | 7.82±3.29 |
| Median (IQR) | | 0.50 (0.00–1.00) | 1.00 (0.50–3.00) | 0.50 (0.50–3.00) | 1.00 (0.50–3.00) | 1.00 (0.50–4.00) | 0.59 (0.00–4.23) | 6.93 (5.16–9.34) |
| 9-month | 9 | 1.33±1.58 | 1.94±1.62 | 1.94±1.62 | 1.88±1.70 | 2.11±1.85 | 2.59±2.92 | 9.41±3.13 |
| Median (IQR) | | 0.50 (0.25–2.75) | 2.00 (0.50–3.75) | 2.00 (0.50–3.75) | 1.50 (0.50–3.75) | 2.00 (0.50–4.25) | 1.37 (0.00–5.90) | 10.03 (6.72–12.18) |
| 1-year | 6 | 1.25±1.57 | 3.16±1.66 | 3.33±1.80 | 3.41±1.65 | 3.41±1.65 | 4.90±3.12 | 13.31±3.61 |
| Median (IQR) | | 2.25 (0.50–4.00) | 3.25 (2.00–4.62) | 3.50 (2.00–5.00) | 3.50 (2.12–5.00) | 3.50 (2.12–5.00) | 4.63 (1.93–8.11) | 13.94 (10.22–15.98) |
| P-value | | 0.029 | 0.007 | 0.009 | 0.009 | 0.007 | 0.025 | 0.001 |
| Pre vs 1-month | | 1.000 | 1.000 | 1.000 | 0.564 | 1.000 | 1.000 | 0.008 |
| Pre vs 3-month | | 0.024 | 0.014 | 0.026 | 0.016 | 0.016 | 0.180 | 0.003 |
| Pre vs 6-month | | 0.018 | 0.011 | 0.011 | 0.007 | 0.007 | 0.018 | 0.003 |
| Pre vs 9-month | | 0.017 | 0.011 | 0.011 | 0.011 | 0.011 | 0.028 | 0.008 |
| Pre vs 1 year | | 0.027 | 0.027 | 0.027 | 0.027 | 0.027 | 0.028 | 0.028 |

*IQR- Interquartile range

**Table 4.  Correlation of age and duration with central corneal sensation and subbasal nerve fiber density (SBNFD) at follow-up visits.**

| Correlation | Central corneal sensation (correlation coefficient) | P-value | SBNFD (correlation coefficient) | P-value |
|---|---|---|---|---|
| Age | | | | |
| 6-month | 0.057 | 0.86 | 0.072 | 0.833 |
| 1-year | 0.00 | 1.00 | -0.086 | 0.872 |
| Disease Duration | | | | |
| 6-month | 0.32 | 0.32 | 0.216 | 0.523 |
| 1-year | -0.016 | 0.976 | -0.185 | 0.725 |

confidence interval (lower limit, upper limit) for SBNFD, SBNFL, central corneal sensation and an average of peripheral corneal sensation at 6 months were (0.12,3.53); (5.61,10.04); (0.09, 2.08); (0.52, 2.59),respectively, and at one year follow up were (1.62,8.17); (9.52,17.09); (-0.39, 2.89); (1.67,5.14), respectively. SBNFD and central corneal sensation in the fellow eyes of enrolled 11 patients were, expressed as mean±SD; median(IQR)- 19.18±6.08;16(16–21) and 5.91±0.30;6 (6–6), respectively. Following corneal neurotization in the affected eyes, there was a statistically significant improvement in SBNFD (p = 0.003) and central corneal sensation (p = 0.003) at 6 months compared with the fellow normal eyes assessed by Wilcoxon Signed Ranks Test.

There was no association between the age of the patient and the duration of the disease with the recovery of corneal sensation and SBNFD (Table 4). On further evaluating the outcomes of corneal neurotization based on preoperative etiology, no significant difference was observed in central corneal sensation improvement (p = 1.000) and increment in SBNFD (p = 0.699) at 6 months. However, corneal sensation recovery observed was significantly better (p = 0.028) in eyes with NK grade 2 compared with NK grade 3, nonetheless, SBNFD was inconsequential (p = 0.575) in the two groups based on disease severity (Table 5).

No intraoperative or postoperative complications related to the surgical techniques were noted during the follow-up. However, two patients (one was of 27 years old with a history of nasal aspergillosis and the other one was 74 years old with no systemic co-morbidity) had a reactivation of viral keratitis 3 and 6 months post-surgery, respectively. Systemic therapeutic doses of antiviral medications with topical steroids were instituted, and patients are being followed-up.

**Table 5.  Effect of corneal neurotization on central corneal sensation and subbasal nerve fiber density at 6-month follow up, in all recruited eyes based on preoperative diagnosis and NK grade severity.**

| Characteristics | Central corneal sensation | | Subbasal nerve fiber density | |
|---|---|---|---|---|
| | (mean±SD) | Median (IQR) | (mean±SD) | Median (IQR) |
| NK grade | | | | |
| Grade 2 | 1.91±1.82 | 1.00 (0.50–4.12) | 2.10±2.56 | 1.07 (0.00–4.73) |
| Grade 3 | 0.40±0.22 | 0.50 (0.25–0.50) | 1.57±2.89 | 0.54 (0.00–3.65) |
| p-value | 0.028 | | 0.575 | |
| Preoperative diagnosis | | | | |
| Herpes | 1.57±1.85 | 0.50 (-.50–4.00) | 1.87±2.42 | 0.78 (0.00–4.23) |
| Facial nerve palsy | 0.75±0.28 | 0.75 (0.50–1.00) | 1.82±3.27 | 0.29 (0.00–5.18) |
| p-value | 1.000 | | 0.699 | |

*IQR- Interquartile range

## Discussion

Corneal neurotisation has evolved as an effective option in the treatment of neurotrophic keratopathy. However, these techniques are not free of predicaments. Our study observed that consolidated donor nerve identification and interpositional nerve graft fascicle separation without collateral damage were the most time-consuming and imperative steps in determining the success of corneal neurotization. Knowledge of anatomical variation and landmarks of donor supratrochlear nerve and sural nerve fascicle separation in the taut position before severing with the use of fibrin glue to secure fascicles around the limbus, perhaps measures to emulate and expedite the surgical procedure. Hence, we attempted these surgical variations in 11 recruited eyes with non-responding neurotrophic keratopathy and evaluated the long-term clinical outcome of the burgeoning corneal neurotization procedure.

In our study, a gradual resolution of corneal clouding with a corresponding significant improvement in visual acuity was observed as early as three months that continued till the last follow-up noted 1-year. However, no general acquiescence to vision improvement has been reported in the literature following corneal neurotization. Leyngold *et* al [17] reported noteworthy vision improvement from 20/70 to 20/20 in one operated case, whereas Jowett and Pineda [18] noticed modest vision improvement in their reported series. In contrast, Benkhatar and colleagues found no improvement in vision [19]. This incongruity can be attributed to variations in the surgical methodology used by different authors, patient selection, disease duration, the structural integrity of the cornea, and pre-existing corneal scarring [10].

At the 1-month follow-up, a statistically significant improvement in ocular surface parameters was observed before visual improvement was perceived. Amelioration in ocular surface parameters, particularly improvement in corneal staining score is pertinent to a significant improvement in NK grade, observed at the 3month clinical examination. Substantial recovery of the ocular surface is closely linked to corneal sensation restoration. Studies have shown that corneal innervation plays a pivotal role in the proliferation of corneal epithelial or limbal stem cells after injury [2]. This is supported by evidence that interactions between the corneal epithelium and corneal innervation upregulate the expression of a5 integrins and E-cadherin, which are necessary for epithelial adhesion to fibronectin in the extracellular matrix and maintain the integrity of the corneal epithelium [20].

In our study, corneal sensation improvement at a statistically significant level was noted at 3 months postoperatively (11/11eyes), consistent with Elbaz et al [10], and Malhotra R et al [21] annotations. Improvements continued for a year after the procedure; however, a longer time course has been reported in earlier studies [22]. A similar observation of corneal sensation improvement at a 3-month follow-up with maximal sensation at 6 months was reported by Kim et al [23] in herpetic NK. However, previous studies [5, 19, 22] and the recently published Rathi et al [24] interim reports observed objective improvement in corneal sensation at 5–6 months. The return of corneal sensation observed was significant in our study, although the absolute value to the contralateral normal cornea was not attained (Fig 5), which is consistent with the results of previous studies [5, 19].

Anatomical evidence of corneal reinnervation on in vivo confocal microscopy is a protracted process that begins with augmentation of corneal subbasal nerve fiber length, detected as early as the 1-month follow-up. Subsequently, the subbasal nerve fiber density became apparent on in vivo confocal microscopy at the 6-month follow-up and progressively increased; however, a linear improvement curve was not established (Fig 6). Our study observed the outcomes harmonized with the findings of Benkhatar et al, confocal microscopic subbasal nerve plexus improvement commenced as early as 3 months, with progressive

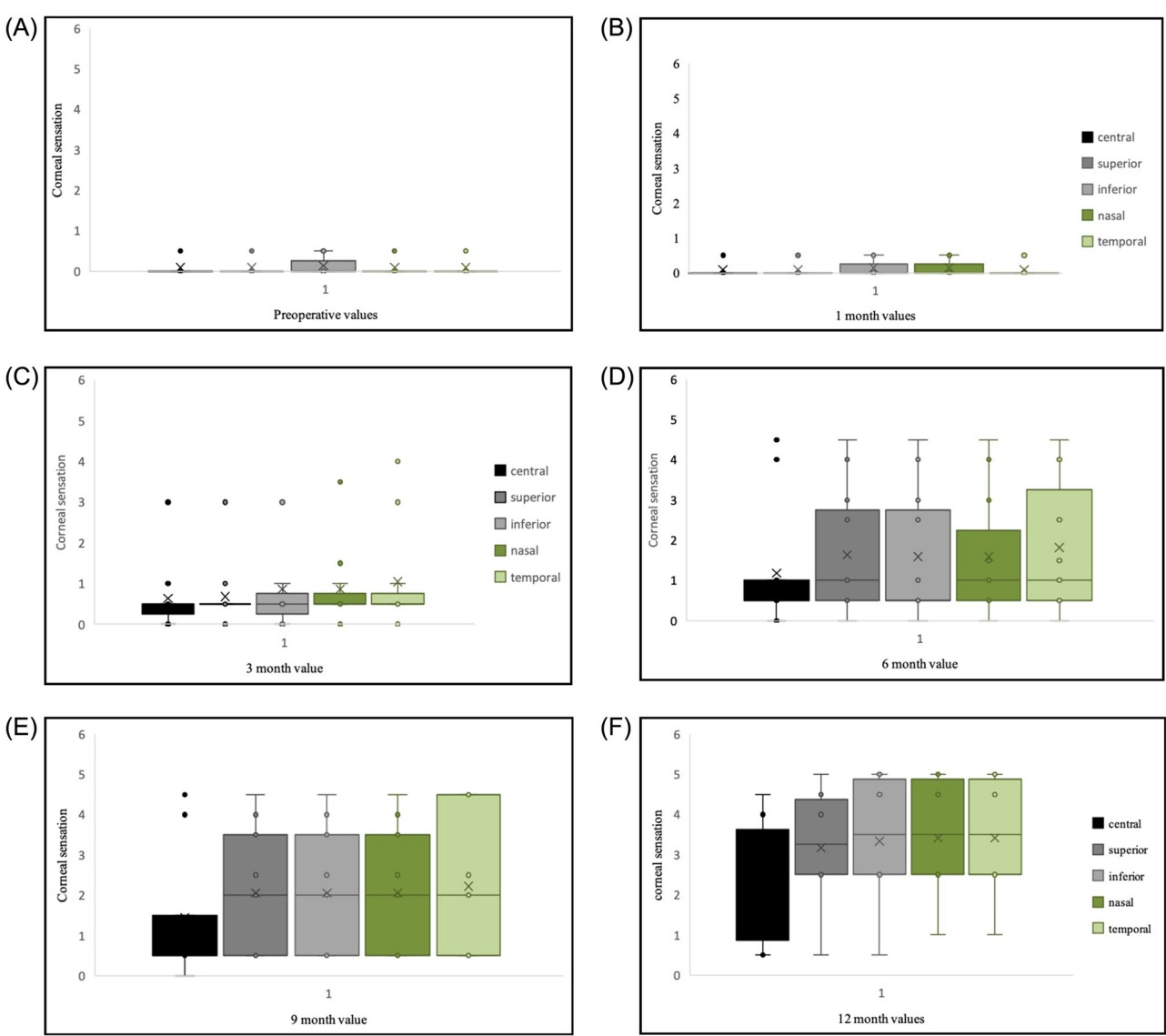

**Fig 5.** A-F. Box and whiskers plots of corneal sensations. It is showing progressive and significant improvement in central and peripheral corneal sensations at postoperative sequential follow-up.

increment over 3–6 months before stabilizing [19]. Our study documented a significant improvement in the subbasal nerve plexus for a year.

Our study speculates an initial increase in sub-basal nerve fiber length, connoting escalation of pre-existing sub-basal nerves and subsequent generation of new nerve bundles over six months and continued at one year (Fig 7). Interestingly, these budding nerve terminals were arranged chaotically and did not follow this pattern. Thus, our study reported that the chronological order of the corneal reinnervation process commenced with an escalation of pre-existing SBNFL spans from 1-month post-surgery, followed by the objective perception of corneal sensation, which spans from 3 months and improvement in SBNFD from 6-month to 1-year following corneal neurotization. The exact mechanism of corneal reinnervation following neurotization is yet to be understood completely because the distal donor nerve fascicles are laid

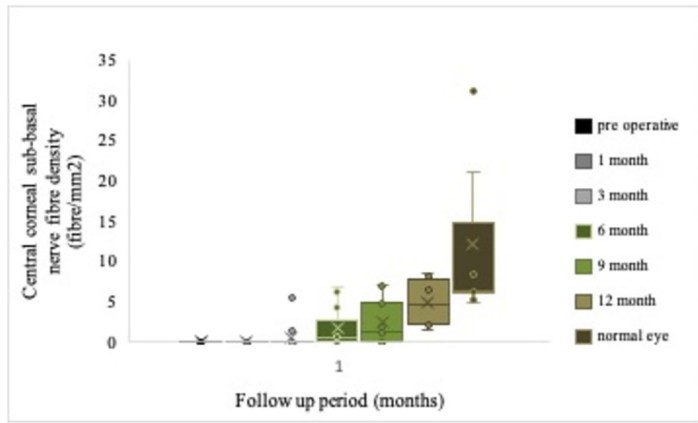

**Fig 6. Box and whiskers plots of subbasal nerve fiber density over serial follow-up.** The graphs depict continuing improvement in subbasal nerve fiber density over serial follow-up after corneal neurotization surgery, though normal contralateral values were not obtained.

around the limbus and not directly coapted to the remaining corneal nerves. By coaptation of the supratrochlear nerve, the new basal laminae of Schwann cells in the donor nerve graft support axonal regeneration that finds its way from the surrounding nerve graft fascicles to the corneal stroma or at least to the subepithelial level, thereby restoring sensation [10].

Despite the promising results of corneal neurotization, two patients with herpes simplex neurotrophic keratopathy developed recurrence in our study. Twenty-seven-year-old male with an antecedent history of paranasal sinus Aspergillus infection developed disciform stromal keratitis at 6 months post-surgery, whereas, a seventy-four-year-old male without systemic illness reported a central epithelial defect three months after the procedure. A review of the literature on the reactivation of herpes simplex keratitis revealed that in addition to cytokines and chemokines, neuropeptide-substance-P, glycoproteins, microRNAs, and other mediators contribute to the pathological immune response of herpes simplex keratitis. All these

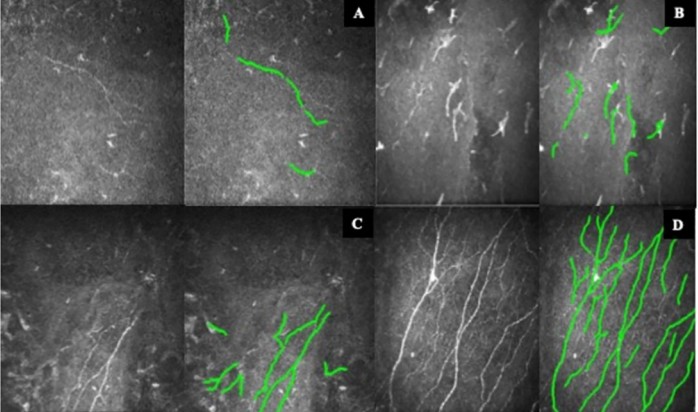

**Fig 7. Representative in vivo confocal microscopy of central corneal subbasal nerve plexus.** (A) Preoperative subbasal nerve fiber plexus, scarcely detectable on ACC Metrics software. (B) One month postoperative, subbasal nerve fiber length became significantly apparent, (C and D) Significant increase in branching and subsequent subbasal nerve fiber density at 6 months and 12 months after the CN surgery.

regulators play dual roles in inhibiting and promoting disease pathogenesis [25]. Therefore, reactivation of the latent virus in the context of hypo immunity (common in both recurrent patients) during the escalation of the sub-basal nerve plexus and consequent release of neuro-peptides resulted in recurrence. However, cellular immunity, inflammatory factors, and molecular assays would help clarify the immunopathological process of reactivation following neurotisation. Both patients continued a therapeutic dose of systemic antiviral medication with topical antibiotic ointment. Topical steroids were also added to the disciform stromal keratitis, and marked improvement in vision from 20/80 to 20/40 with resolution in stromal keratitis was observed in young patient, 5-months after initiation of the treatment; however, older patients showed no improvement in vision with the resolution of the epithelial defect observed at 7-months.

In the literature, young patients have been associated with faster and more complete recovery [26]. However, no discernible correlation was noted between age and improvement in corneal sensation and subbasal nerve plexus parameters. Similarly, no significant difference was perceived between herpes etiology NK and facial nerve palsy associated with NK on central corneal sensation and SBNFD. In the literature, there is a dearth of evidence regarding the comparison of two different etiology-related outcomes. Perhaps, a large sample size would have helped establish a correlation between SBNFD and disease duration and different etiology associated NK. While more significant improvement in corneal sensation was observed in NK grade 2 compared to NK grade 3 eyes, at non-significant SBNFD increment. The possible explanation was that the analogous regenerating SBNFD was only perceptible sufficient to reinnervate the less stromal depth involving cornea than deeper corneal involving NK stage.

Complications at the donor site related to sural nerve graft harvestings, such as loss of sensation, discomfort, and allodynia in the lower leg or foot [27] were anticipated; however, none of these complications were evident at three months postoperative period. The elevated cost of the procedure and multidisciplinary approach are the major limitations to the easy adoption of this novel technique [28].

## Conclusion

In summary, our study demonstrated the efficacy of corneal neurotization in treating the underlying pathology of neurotrophic keratopathy and substantiated the routine practice of this technique by simplifying the intricacies observed during the procedure.

## Supporting information

**S1 Table. Study's underlying data.** Data of the enrolled patients in the study.
(DOCX)

**S2 Table. TREND statement checklist.** Checklist of our intervention study.
(DOCX)

**S1 File. Study's protocol.** Proposed protocol for the ethical approval of our study titled Clinical outcomes of corneal neurotization using sural nerve graft in neurotrophic keratopathy.
(DOC)

## Acknowledgments

Postgraduate Institute of Medical Education and Research, Chandigarh-160012, India.

## Author Contributions

**Conceptualization:** Manu Saini, Aman Kalia, Arun K. Jain, Sunil Gaba.

**Data curation:** Chintan Malhotra, Amit Gupta, Tanvi Soni, Kulbhushan Saini, Parul Chawla Gupta, Manpreet Singh.

**Formal analysis:** Chintan Malhotra, Amit Gupta, Tanvi Soni, Kulbhushan Saini, Manpreet Singh.

**Investigation:** Aman Kalia, Arun K. Jain, Kulbhushan Saini.

**Methodology:** Manu Saini, Sunil Gaba, Tanvi Soni.

**Supervision:** Arun K. Jain, Chintan Malhotra, Amit Gupta.

**Validation:** Amit Gupta, Manpreet Singh.

**Visualization:** Arun K. Jain, Kulbhushan Saini, Parul Chawla Gupta, Manpreet Singh.

**Writing – original draft:** Manu Saini.

**Writing – review & editing:** Aman Kalia, Arun K. Jain, Sunil Gaba, Parul Chawla Gupta.

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
