## [Decision Letter · Decision Letter 0]

31 May 2023

PONE-D-23-06245Clinical Outcomes of Corneal Neurotization Using Sural Nerve Graft in Neurotrophic KeratopathyPLOS ONE

Dear Dr. Jain,

Thank you for submitting your manuscript to PLOS ONE. After careful consideration, we feel that it has merit but does not fully meet PLOS ONE’s publication criteria as it currently stands. Therefore, we invite you to submit a revised version of the manuscript that addresses the points raised during the review process.

We look forward to receiving your revised manuscript.

Kind regards,

Bhavana Sharma, MS

Academic Editor

PLOS ONE

Journal Requirements:

3. Please amend your manuscript to include your abstract after the title page.

Additional Editor Comments:

Dear Authors

Please address the enclosed comments of reviewers , following which the MS would be re-evaluated for adequacy of publication .Furthermore kindly give your comments on the following:

1.Please specify the severity grading in each diagnosis subgroup .How was homogeneity ensured wrt clinical features and severity grade in each group .

2. Was the adjunctive medication pre/postop ,uniform in all pts/study groups?

3.What preoperative treatment was accorded to each group and what was the follow up period ? whether it was uniform for all patients ,before subjecting them to study intervention. Postoperative medication , dose/ duration would be relative to the same . 

4.Was the effect of neurotization dependent on preoperative diagnosis, severity and supportive medication .If it was better in any particular group/patients ,it can be included in discussion along with pathophysiological correlation .

5.There remains a possibility of disease recurrence in HSV /HZO, how was this ruled out and adequacy of medication ensured  before subjecting the patient for neurotization .

Reviewers' comments:

Reviewer's Responses to Questions

**Comments to the Author**

1. Is the manuscript technically sound, and do the data support the conclusions?

Reviewer #1: Partly

Reviewer #2: Yes

Reviewer #3: Yes

2. Has the statistical analysis been performed appropriately and rigorously? 

Reviewer #1: No

Reviewer #2: Yes

Reviewer #3: No

3. Have the authors made all data underlying the findings in their manuscript fully available?

Reviewer #1: Yes

Reviewer #2: Yes

Reviewer #3: Yes

4. Is the manuscript presented in an intelligible fashion and written in standard English?

Reviewer #1: Yes

Reviewer #2: No

Reviewer #3: Yes

5. Review Comments to the Author

Reviewer #1: Sample size should be calculated based on the study design and test hypothesis, especially to provide sufficient power to demonstrate efficacy. It should not be based on recruitment availability.

The statistical methods need more work. For repeated measures, a repeated measure model should be used. Where was the Fisher’s exact test used?

With a small, limited sample size, there are many outcomes and tests to consider. P-values should be adjusted. Why was the significance level set at 0.01 for the correlation, but 0.05 for other tests?

The subtitle “summary background data” in the abstract is not correct.

In Figure 1, the flowchart should not be separated into two arms after follow-up.

The method used to compare the normal group is not described.

The footnote in Table 4 can be omitted.

Reviewer #2: The authors cover a very important and interesting topic. Their results are impressive and highlight a very promising emerging treatment for neurotrophic keratopathy. The intraoperative and slit lamp images are beautiful, showcasing the potential impact for patients. The study is not unique since there have been several other similar case series, but given the still relative novelty of this treatment and the impressive results, I do think it merits publication. A few points for improvement:

1. There are numerous grammatical errors, incorrect usage of words, and awkward sentences. I recommend professional copyediting to be performed prior to publication.

2. The authors simply state that medical management had failed, but there are not enough details on what kind of medical management. Please clarify what medical treatments were used and for how long prior to surgery.

3. The authors detailed the post-operative medications prescribed to all patients, but it is not clear if other medical treatments for NK were allowed (i.e. artificial tears, serum tears, scleral lenses, etc).

4. Was there any significant difference in the results based on the etiology of NK? It may be difficult due to the already small sample size, but it would be interesting to know if certain diagnoses portend a better prognosis.

Reviewer #3: Comments to the Author

1. Overall, the manuscript requires major revision towards statistical analysis. Also, I have a great concern about 1 year conclusions because of great loss to follow-up (even 46%, 6 out of 11 patients/eyes). I recommend ending reporting with the 9th month measurement, because of the reliability of results and its potential for generalisation.

2. Introduction:

Line 47-49 – There is an explanation about evidence of neuroparalytic-like corenal alterations in mice, but it is necessary to explain this state in humans a bit. Please add an explanation and an appropriate reference.

Line 55-56 – References regarding high costs missed. Please add.

3. Methods:

Line 175-180 – Did the authors evaluate the normality of numerical data? Which of the methods did you use? Please, include this in statistical analysis subsection of the methods.

Also, report full names of applied statistical tests for comparing values of evaluated parameters within the statistical methods and in the description of all tables where appropriate. Insert symbols like €, £, etc. where more than one test was applied.

Please, explain why did you choose the significance level of 0.01 for correlation analysis. It will not change the conclusion if it is 0.05, as it is recommended for biology research.

4. Results:

Correct capital P value into small letter p value throw-out whole manuscript and in tables also.

P value can not be 0.00, so change it into number with 3 decimals. If it is 0.000 in the statistical software, report it as p<0.001.

Round all p values on 3 decimals, even when the third decimal is 0, in the whole manuscript and tables also.

Table 1. Please correct capital E into small letter e in the second column “Neurotrophic keratopathy eyes”

Table 2. Report p value for comparing all measurements pre, 1 month, 3 months, 6 months, 9 months, and 12 months for NK grade.

Line 220-221: Commas missed in the sentence: “…(0.52, 2.59), respectively, and at one year follow up were (1.62,8.17); (9.52,17.09); (-0.39, 2.89); (1.67,5.14), respectively”. Please add.

Line 230: bracket missed in the sentence: “However, two patients (one was of 27-year old with a history of nasal aspergillosis and the other one was 74-year-old with no systemic co-morbidity) had reactivation of viral keratitis 3 and 6 months post-surgery, respectively.”

Discussion:

Please remove p values obtained in your research from this section and reword sentences with p values within to fit well the rest of the text.

Figure 5. I am concerned about the resolution of this figure. Please make it better. It is unreadable now. Instead of making one, submit 5 figures separately.

Figure 5. Please enlarge the resolution of this figure. It is not good enough.

6. PLOS authors have the option to publish the peer review history of their article (what does this mean?). If published, this will include your full peer review and any attached files.

Reviewer #1: No

Reviewer #2: No

Reviewer #3: No

---

## [Author Response · Author response to Decision Letter 0]

12 Jul 2023

Dear Editor-in-chief,

PLOS ONE 

Thank you for giving us the opportunity to submit a revised manuscript titled-Clinical outcome of corneal neurotization using sural nerve graft in neurotrophic keratopathy with manuscript number- PONE-D-23-06245

We appreciate the time and effort that you and the reviewers have dedicated to providing your valuable feedback on our manuscript. We are grateful to the reviewers for their insightful comments on our paper. We have been able to incorporate changes to reflect the suggestions provided by you and the reviewers. We have highlighted the changes within the manuscript. Here is a point-by-point response to the reviewers’ comments and concerns 

Journal Requirements:

Response: Your feedbacks are valuable to us, and we have made the specified changes in the revised manuscript to meet the PLOS ONE's style, required for publication. 

2. In your Data Availability statement, you have not specified where the minimal data set underlying the results described in your manuscript can be found.

Response: We are extremely sorry to hear about that and we appreciate the higher effort that you have exerted. In revised submission, minimal underlying data set as has been uploaded in the Supporting Information files.

The file name and captions are as follows- S1 Table. Study’s underlying data. Data of the enrolled patients in the study. 

3. Please amend your manuscript to include your abstract after the title page.

Response: We thank you for your accommodating remark. We have inserted the abstract after the title page as per your suggestion.

4. Please include captions for your Supporting Information files at the end of your manuscript, and update any in-text citations to match accordingly

Response: Your feedback is much appreciated. The captions of the supporting information files have been added at the end of revised manuscript. 

Editor 

Thank you for the interest in our manuscript. We highly appreciate the Editor for the considerable efforts in the manuscript. We have revised the manuscript as recommended by you. We hope to hear from you shortly with a positive conclusion.

ADDITIONAL EDITOR COMMENTS

1.Please specify the severity grading in each diagnosis subgroup. How was homogeneity ensured wrt clinical features and severity grade in each group. 

Response: We would like to express our gratitude for your thoughtful suggestion and comments. As per your suggestion, the severity grade of neurotrophic keratopathy (NK grade) has been added against the herpes simplex, herpes zoster and facial nerve palsy patients shown in Table 1 in yellow high lightened. 

In our 11 unilateral neurotrophic keratopathy patients of herpes simplex, herpes zoster and facial nerve palsy associated with neurological insult were recruited after excluding associated lid malposition, prior corneal surgery, history of diabetes, leprosy or peripheral neuropathy. Mickie’s classification for the grading of neurotrophic keratopathy which is a persistent epithelial defect representing stage-2 and corneal ulceration with stromal involvement demonstrating stage–3 was used to maintain consistency concerning clinical features and severity grade. However, because of the small sample size equal segregation of patients and group division in respect of NK grade, and disease etiology was not feasible. Nevertheless, as per your much-appreciated suggestion, we attempt to study the effect of corneal neurotization based on preoperative diagnosis and NK grade severity. 

2. Was the adjunctive medication pre/postop ,uniform in all pts/study groups?

Response: Thank you for pointing this out. Preoperatively all the recruited patients were on preservative-free lubricating eye drops and ointment for 6 months. Systemic oral tetracycline in a dose of 100mg BD for two weeks was considered initially at diagnosis for stage 2 and 3 NK grades eyes. The use of topical antibiotic eye drops was considered intermittently to prevent secondary infection. It has been added in the methodology section in yellow highlight (page no-6). None of the patients used the autologous serum, contact lens, punctal occlusion, tenon patch graft, amniotic membrane graft, any corneal surgical intervention or tarsorrhaphy. 

 Postoperative systemic antibiotics for seven days, topical steroid drops in the tapering dose, topical antibiotics, lubricating eye drops, and ointment were prescribed for 6 weeks in all recruited eyes. The antiviral drugs in therapeutic doses were prescribed till the patient was on the topical steroids (6 weeks), followed by a prophylactic antiviral dose for a period of a total of six months to prevent reactivation of viral keratitis in viral etiology NK. The same has been incorporated in the manuscript under the surgical procedure section in the last paragraph (page no-9).

3.What preoperative treatment was accorded to each group and what was the follow up period ? whether it was uniform for all patients ,before subjecting them to study intervention. Postoperative medication, dose/ duration would be relative to the same. 

Response: We highly appreciate your comments. All the recruited patients were on preservative-free lubricating eye drops and ointment for at least 6 months, before enrolling them for the surgical procedure. In addition, systemic oral tetracycline in a dose of 100mg BD for two weeks was considered initially at diagnosis for stage 2 and 3 NK patients. The use of topical antibiotic eye drops to prevent secondary infection at NK grades 2 and 3 was considered intermittently. (page no -6). Hence, it was uniform for all the recruited patients.

Postoperative systemic antibiotics for seven days, topical steroid drops in the tapering dose, topical antibiotics (4 times a day), preservative-free lubricating eye drops (4 times a day), and ointment (3 times a day) were prescribed for 6 weeks in all recruited eyes. The antiviral drugs in therapeutic dose (Acyclovir 400mg 5 times a day) were prescribed till the patient was on the topical steroids (6 weeks), followed by prophylactic antiviral dose (Acyclovir 400mg 2 times a day) for a period of total six months to prevent reactivation of viral keratitis in viral etiology NK (page no-9). 

4.Was the effect of neurotization dependent on preoperative diagnosis, severity and supportive medication. If it was better in any particular group/patients, it can be included in discussion along with pathophysiological correlation.

Response: We would like to express our gratitude for your thoughtful suggestion to study the effect of neurotization dependent on preoperative diagnosis, severity and supportive medication. Hence, on evaluating the outcomes of corneal neurotization based on preoperative etiology, no significant difference was observed in central corneal sensation improvement (p=1.000) and increment in SBNFD (p=0.699) at 6 months. However, corneal sensation recovery observed was significantly better (p=0.028) in eyes with NK grade 2 compared with NK grade 3, nonetheless, SBNFD was inconsequential (p=0.575) in the two groups based on disease severity. The evidence is supported by the provided Table 5. The same has been included in the result section (page no-15) with possible explanation in the discussion part (page no-20) of the revised manuscript. 

5.There remains a possibility of disease recurrence in HSV /HZO, how was this ruled out and adequacy of medication ensured before subjecting the patient for neurotization.

Response: Thank you for the thoughtful study of our material and insightful comment. We completely agreed that there was a possibility of disease recurrence in HSV /HZO, therefore, a therapeutic dose of antiviral drugs-acyclovir 400mg 5 times a day postoperatively was prescribed till the patient was on the topical steroids (6 weeks), followed by prophylactic antiviral dose (400mg BD) for a period of a total of six months in viral etiology NK to obviate reactivation of viral keratitis (page no-9). Adherence was ensured by confirming through the medication strips at every follow-up and ensuring the dosing, was as per instructions by the patient and relatives themselves amplified by phone calls before follow-up scheduled dates. 

Reviewer #1

We highly appreciate your attentive study of our manuscript and insightful remarks, which have greatly helped in the refinement of the document. We have revised the manuscript as recommended by you. We are looking forward to hearing your decision soon.

 REVIEWERS' COMMENTS

Sample size should be calculated based on the study design and test hypothesis, especially to provide sufficient power to demonstrate efficacy. It should not be based on recruitment availability.

Response: We appreciate your attentive study of our material and insightful remarks, which helped us greatly in the refinement of our manuscript. The epidemiological data of NK are scarcely reported in the literature, the prevalence and incidence of NK was estimated as 1.6/10,000 from the epidemiological data on conditions associated with NK- herpetic keratitis and post-surgical procedures1. Therefore, the above-mentioned value of epidemiological data was used to calculate the sample size with a 10% margin of error. 

The following formula was used for the sample size calculation n = N*X / (X + N – 1), where, X = Zα/22 ¬*p*(1-p) / MOE2, and Zα/2 is the critical value of the Normal distribution at α/2 (e.g. for a confidence level of 99%, and α is 0.10), MOE is the margin of error, p is the sample proportion, and N is the population size. The same has been incorporated in the manuscript (page no-10,11). 

 1Sacchetti M, Lambiase A. Diagnosis and management of neurotrophic keratitis. Clin Ophthalmol. 2014 Mar 19;8:571-9. doi: 10.2147/OPTH.S45921.

The statistical methods need more work. For repeated measures, a repeated measure model should be used. Where was the Fisher’s exact test used?

Response: Your feedback is much appreciated. The statistical methods have been revised and names of applied statistical tests for comparing the clinical parameters have been added in the statistical section. Because of small sample size and only 6 eyes completed 1year follow-up, serve as a limitation for using repeated measure tests. Fisher’s exact test was used for assessing NK grade improvement at follow-up visits, being categorical variable.

With a small, limited sample size, there are many outcomes and tests to consider. P-values should be adjusted. Why was the significance level set at 0.01 for the correlation, but 0.05 for other tests?

Response: We are extremely sorry to hear about that and we appreciate the higher effort that you have exerted. The correlation was calculated by SPSS software version 20 which was calculated at 0.05 as well as 0.01, however, it showed the highest level and the correlation perceived was non-significant. Therefore, a significance level of 0.01 for the correlation has been omitted and the same has been rectified in the revised submission.

The subtitle “summary background data” in the abstract is not correct.

Response: Thank you for your comments. Indeed, the subtitle “summary background data” in the abstract is not correct. Therefore, we have just deleted the subtitle from the abstract. 

In Figure 1, the flowchart should not be separated into two arms after follow-up.

Response: Thank you for your valuable comments. As per your suggestion, two arms after follow-up in the flow chart have been deleted. 

The method used to compare the normal group is not described.

Response: We highly appreciate your efforts and insight. As per your suggestion SBNFD and central corneal sensation were compared with the fellow eyes of the recruited patients using Wilcoxon Signed Ranks Test. There was a statistically significant improvement in SBNFD (p=0.003) and central corneal sensation (p=0.003) at 6 months compared with the fellow normal eyes. The same has been added in the result section in highlight (page no-15). 

The footnote in Table 4 can be omitted.

Response: Thank you for your guidance. The footnote in the Table 4 has been omitted

Reviewer #2 

REVIEWERS' COMMENTS

1. There are numerous grammatical errors, incorrect usage of words, and awkward sentences. I recommend professional copyediting to be performed prior to publication.

Response: Thank you for your careful checks and corrections. English grammar editing has been performed using online available software before uploading the revised manuscript for publication. 

2. The authors simply state that medical management had failed, but there are not enough details on what kind of medical management. Please clarify what medical treatments were used and for how long prior to surgery.

Response: Your feedback is much appreciated, and helped in the refinement of our manuscript. All the recruited eyes were on preservative-free artificial tears in the form of drops and ointment at all stages of disease severity for at least 6 months before enrolment. In addition, systemic oral tetracycline in a dose of 100mg BD for two weeks was considered initially at diagnosis for stage 2 and 3 NK grades before proclaiming them medically non-respondent eyes. The use of topical antibiotic eye drops to prevent infection at NK grade 2 and 3 were considered intermittently. In the case of herpes etiology NK, patients did not have a recurrence in the past six months, before enrolment. It has been included in the study design and participants sub-section (page no-6).

3. The authors detailed the post-operative medications prescribed to all patients, but it is not clear if other medical treatments for NK were allowed (i.e. artificial tears, serum tears, scleral lenses, etc).

Response: Thank you for your valuable comment. None of the recruited patients used the autologous serum, contact lens, punctal occlusion, tenon patch graft, amniotic membrane graft, any corneal surgical intervention or tarsorrhaphy. 

4. Was there any significant difference in the results based on the etiology of NK? It may be difficult due to the already small sample size, but it would be interesting to know if certain diagnoses portend a better prognosis.

Response: We would like to express our gratitude for your thoughtful suggestion to study the effect of neurotization based on etiology of NK. Hence, on evaluating the outcomes of corneal neurotization based on etiology, no significant difference was observed in central corneal sensation improvement (p=1.000) and increment in SBNFD (p=0.699) at 6 months. The evidence is supported by the provided Table 5. The same has been included in the result (page no-15) and discussion section (page no-20) of the revised manuscript. 

Reviewer #3 

REVIEWERS' COMMENTS

1. Overall, the manuscript requires major revision towards statistical analysis. Also, I have a great concern about 1 year conclusions because of great loss to follow-up (even 46%, 6 out of 11 patients/eyes). I recommend ending reporting with the 9th month measurement, because of the reliability of results and its potential for generalisation.

Response: We would like to express our gratitude for your thoughtful suggestions. The statistical analysis has been rectified in the revised manuscript and names of applied statistical tests for comparing the clinical parameters have been added. 

Regarding one year follow-up, the objective improvement of corneal sensation documented in the literature was around 5–6 months. However, improvement noted was continued for about a year after the procedure, though a longer time course has been reported in the earlier studies.1,2 Hence to ensure the maximal improvement in corneal sensation, though in small sample size one-year follow-up was preferred. We completely agree with the reviewer regarding small sample size that serves as limitation. We have included references (below) to support our conclusions. 

1Terzis JK, Dryer MM, Bodner BI. Corneal neurotization: a novel solution to neurotrophic keratopathy. Plast Reconstr Surg 2009;123:112–20 

2Jacinto F, Espana E, Padilla M, Ahmad A, Leyngold I. Ipsilateral supraorbital nerve transfer in a case of recalcitrant neurotrophic keratopathy with an intact ipsilateral frontal nerve: a novel surgical technique. Am J Ophthalmol Case Rep 2016;4:14–7. 

2. Introduction:

Line 47-49 – There is an explanation about evidence of neuroparalytic-like corneal alterations in mice, but it is necessary to explain this state in humans a bit. Please add an explanation and an appropriate reference

Response: We highly appreciate your insight and acknowledge your contributions to improving our manuscript. As per your suggestions, neurotrophic keratopathy pathophysiology recognized in humans has been added in the introduction section (page no-4) with corresponding reference in yellow highlight. 

Line 55-56 – References regarding high costs missed. Please add.

Response: Your feedback is much appreciated. As per your valuable comment with reference regarding high costs have been added (Reference no-28) as limitation of the procedure (page no-21). 

3. Methods:

Line 175-180 – Did the authors evaluate the normality of numerical data? Which of the methods did you use? Please, include this in statistical analysis subsection of the methods.

 Response: Thank you for your insightful observation regarding the evaluation of the normality of data. Our clinical data did not exhibit normality, assessed by Shapiro–Wilk test, therefore non parametric tests were performed for analysis. The same has been incorporated in the statistical section of the revised manuscript. 

Also, report full names of applied statistical tests for comparing values of evaluated parameters within the statistical methods and in the description of all tables where appropriate. Insert symbols like €, £, etc. where more than one test was applied.

Response: The names of applied statistical tests for comparing the clinical parameters (preoperative versus postoperative) at defined follow-up have been added in the subsection of statistical analysis. We believe it would provide a more accurate representation of our findings

Please, explain why did you choose the significance level of 0.01 for correlation analysis. It will not change the conclusion if it is 0.05, as it is recommended for biology research.

Response: We are extremely sorry to hear about that and we appreciate the higher effort that you have exerted. The correlation was calculated by SPSS software version 20 which was calculated at 0.05 as well as 0.01, however, it showed the highest level and the correlation perceived was non-significant. Therefore, a significance level of 0.01 for the correlation has been omitted and the same has been rectified in the revised submission.

4. Results:

Correct capital P value into small letter p value throw-out whole manuscript and in tables also.

Response: We are grateful for your input and suggestions. Capital P value has been replaced with small p value in the manuscript and tables as well. 

P value cannot be 0.00, so change it into number with 3 decimals. If it is 0.000 in the statistical software, report it as p<0.001.

Response: We highly appreciate your comments that would help in improving the quality of our manuscript and therefore specified changes have been included in the manuscript and accordingly high lightened. 

Round all p values on 3 decimals, even when the third decimal is 0, in the whole manuscript and tables also.

Response: Thank you for bringing attention to round all p values on 3 decibels. As per your suggestion changes have been incorporated in the revised manuscript, including tables. 

Table 1. Please correct capital E into small letter e in the second column “Neurotrophic keratopathy eyes”

Response: Thank you for your valuable comments. As per your suggestion, capital E in “Neurotrophic keratopathy eyes” has been substituted with small ‘e’. 

Table 2. Report p value for comparing all measurements pre, 1 month, 3 months, 6 months, 9 months, and 12 months for NK grade.

Response: We appreciate your thoughtful evaluation of our manuscript. Following your suggestion p value for NK grade has been added in the Table 2 and high lightened in the revised file. 

Line 220-221: Commas missed in the sentence: “…(0.52, 2.59), respectively, and at one year follow up were (1.62,8.17); (9.52,17.09); (-0.39, 2.89); (1.67,5.14), respectively”. Please add.

Response: Thank you for your meticulous check and remark, as per your suggestion commas has been added and high lightened in the revised manuscript. 

Line 230: bracket missed in the sentence: “However, two patients (one was of 27-year old with a history of nasal aspergillosis and the other one was 74-year-old with no systemic co-morbidity) had reactivation of viral keratitis 3 and 6 months post-surgery, respectively.”

Response: Thank you for your careful checks and corrections. Bracket has been incorporated as per your guidance. 

Discussion:

Please remove p values obtained in your research from this section and reword sentences with p values within to fit well the rest of the text.

Response: Thank you for your valuable comments. As per your suggestion, p values of the clinical parameters have been removed from the discussion section. 

Figure 5. I am concerned about the resolution of this figure. Please make it better. It is unreadable now. Instead of making one, submit 5 figures separately.

Figure 5. Please enlarge the resolution of this figure. It is not good enough.

Response: We are grateful for your input and suggestions. Figure 5 shows a collage of corneal sensation improvement plots, have been submitted separately in the revised manuscript, amenable to reading.

 We appreciate for Editors/Reviewers’ warm work earnestly and hope that the corrections will meet with approval. We believe that these modifications address the concern raised and contribute to the overall strength of our paper. We look forward to hearing from you at your earliest convenience and to respond to any further questions and comments you may have. 

Anticipating positive response. 

Sincerely

Prof. Arun K. Jain

Professor and Head of Cornea, Cataract and Refractive Surgery

Advanced Eye Centre, Post Graduate Institute of Medical Education and

Research, Chandigarh-160012, India.

---

## [Decision Letter · Decision Letter 1]

7 Aug 2023

PONE-D-23-06245R1Clinical Outcomes of Corneal Neurotization Using Sural Nerve Graft in Neurotrophic KeratopathyPLOS ONE

Dear Dr. Jain,

Thank you for submitting your manuscript to PLOS ONE. After careful consideration, we feel that it has merit but does not fully meet PLOS ONE’s publication criteria as it currently stands. Therefore, we invite you to submit a revised version of the manuscript that addresses the points raised during the review process.

We look forward to receiving your revised manuscript.

Kind regards,

Bhavana Sharma, MS

Academic Editor

PLOS ONE

Journal Requirements:

Additional Editor Comments:

Authors are requested further , to do necessary corrections/modifications as desired by reviewers .

Reviewers' comments:

Reviewer's Responses to Questions

**Comments to the Author**

1. If the authors have adequately addressed your comments raised in a previous round of review and you feel that this manuscript is now acceptable for publication, you may indicate that here to bypass the “Comments to the Author” section, enter your conflict of interest statement in the “Confidential to Editor” section, and submit your "Accept" recommendation.

Reviewer #1: (No Response)

Reviewer #3: (No Response)

2. Is the manuscript technically sound, and do the data support the conclusions?

Reviewer #1: Yes

Reviewer #3: Yes

3. Has the statistical analysis been performed appropriately and rigorously? 

Reviewer #1: Yes

Reviewer #3: No

4. Have the authors made all data underlying the findings in their manuscript fully available?

Reviewer #1: Yes

Reviewer #3: Yes

5. Is the manuscript presented in an intelligible fashion and written in standard English?

Reviewer #1: Yes

Reviewer #3: Yes

6. Review Comments to the Author

Reviewer #1: Table 3 better report median (IQR or min max) since nonparametric method was used.

Flowchart, there are still two arms. This is one-arm study. Two-arm chart usually refers to two treatment groups.

The one year follow up box should be put underneath the 6 month box in one line.

Reviewer #3: I thank the authors for their revised manuscript. They did an extensive rework and they achieved a significant improvement, but there is more that can be done in order to improve the quality of this manuscript.

The main requirement is regarding the statistical subsection. Fisher`s exact test is recommended for categorical data, and NK grade severity is ordinal, so the more suitable statistical method in the before-after empirical situation is Wilcoxon signed rank test. So, there is need to reword the part of statistical analysis that talks about the usage of Wilcoxon signed rank test like this: “Wilcoxon signed rank test was used for …” and to change the results regarding NK grade severity and follow-up.

7. PLOS authors have the option to publish the peer review history of their article (what does this mean?). If published, this will include your full peer review and any attached files.

Reviewer #1: No

Reviewer #3: No

---

## [Author Response · Author response to Decision Letter 1]

23 Aug 2023

Dear Editor-in-chief,

PLOS ONE

 We appreciate the thorough review of our manuscript PONE-D-23-06245R1 titled "Clinical Outcomes of Corneal Neurotization Using Sural Nerve Graft in Neurotrophic Keratopathy " and the valuable feedback provided by the editor and reviewers.

We believe that these revisions enhance the overall quality and accuracy of our manuscript

JOURNAL REQUIREMENTS:

Response- We have meticulously reviewed our reference list to ensure its completeness and accuracy. Reference 4 and 6, which were not identified in the PubMed search have been replaced with the relevant and appropriate citations. 

REVIEW COMMENTS TO THE AUTHOR 

Reviewer #1

We sincerely appreciate your thorough review of our manuscript. Your feedback has been invaluable in improving the quality of our work. We have carefully considered your comments and made the necessary revisions accordingly. 

Comment-: Table 3 better report median (IQR or min max) since nonparametric method was used. Flowchart, there are still two arms. This is one-arm study. Two-arm chart usually refers to two treatment groups.

The one year follow up box should be put underneath the 6 month box in one line.

Response- We are grateful for your thoughtful input, which has undoubtedly improved the quality of our manuscript. We have updated the table 3 to report the median along with the interquartile range (IQR) to provide a more accurate representation of the data.

We acknowledge the confusion created by the two-arm chart representation. Therefore, we have corrected the error and updated the flowchart to clearly reflect the one-arm study design, eliminating any ambiguity in the visual presentation 

Thank you for pointing out the arrangement of the timeline boxes. We have rearranged the flow chart follow up depiction, one-year follow-up box beneath the 6-month box to ensure that the flowchart now accurately represents the temporal sequence of the study.

Reviewer #3:

We truly appreciate your diligence in reviewing our manuscript and your insightful comment on the statistical analysis section. Your expertise has been invaluable in identifying the appropriate statistical method for our study. We believe that these changes have strengthened the overall presentation and interpretation of our study findings.

Comment- The main requirement is regarding the statistical subsection. Fisher`s exact test is recommended for categorical data, and NK grade severity is ordinal, so the more suitable statistical method in the before-after empirical situation is Wilcoxon signed rank test. So, there is need to reword the part of statistical analysis that talks about the usage of Wilcoxon signed rank test like this: “Wilcoxon signed rank test was used for …” and to change the results regarding NK grade severity and follow-up.

Response- In response to your suggestion, we have carefully considered the statistical analysis and used the Wilcoxon signed rank test for before and after corneal neurotization to asses NK grade severity. 

Consequently, we have reworded the statistical analysis section to accurately reflect the utilization of the Wilcoxon signed rank test for assessing the changes in NK grade severity. 

Furthermore, we have recalculated and updated the results related to NK grade severity and follow-up in table 2 (Yellow highlighted). This modification ensures the statistical integrity and consistency of our study.

We are grateful for your meticulous attention to detail and your guidance in enhancing the accuracy of our research. Your insights have contributed significantly to the overall quality of the manuscript. 

Thank you for your time and commitment to improving our work.

Sincerely 

Prof. Arun K. Jain

Professor and Head of Cornea, Cataract and Refractive Surgery

Advanced Eye Centre, Post Graduate Institute of Medical Education and

Research, Chandigarh-160012, India.

---

## [Decision Letter · Decision Letter 2]

9 Nov 2023

Clinical Outcomes of Corneal Neurotization Using Sural Nerve Graft in Neurotrophic Keratopathy

PONE-D-23-06245R2

Dear Dr. Jain

We’re pleased to inform you that your manuscript has been judged scientifically suitable for publication and will be formally accepted for publication once it meets all outstanding technical requirements.

Kind regards,

Bhavana Sharma, MS MAMS

Academic Editor

PLOS ONE

Reviewer #1: No

---

## [Editor Report · Acceptance letter]

14 Nov 2023

PONE-D-23-06245R2 

Clinical Outcomes of Corneal Neurotization Using Sural Nerve Graft in Neurotrophic Keratopathy 

Dear Dr. Jain:

I'm pleased to inform you that your manuscript has been deemed suitable for publication in PLOS ONE. Congratulations! Your manuscript is now with our production department. 

Kind regards, 

on behalf of

Dr. Bhavana Sharma 

Academic Editor

PLOS ONE